# A Study on the Planarian Model Confirms the Antioxidant Properties of Tameron against X-ray- and Menadione-Induced Oxidative Stress

**DOI:** 10.3390/antiox12040953

**Published:** 2023-04-18

**Authors:** Elena Tsarkova, Kristina Filippova, Vera Afanasyeva, Olga Ermakova, Anastasia Kolotova, Artem Blagodatski, Artem Ermakov

**Affiliations:** 1Institute of Theoretical and Experimental Biophysics of the Russian Academy of Sciences, 142290 Pushchino, Moscow Region, Russia; 2ANO Engineering Physics Institute, Bolshoi Udarny Pereulok, 142210 Serpukhov, Moscow Region, Russia

**Keywords:** planarians, Tameron, antioxidant, regeneration, radioprotection

## Abstract

Ionizing radiation and radiation-related oxidative stress are two important factors responsible for the death of actively proliferating cells, thus drastically reducing the regeneration capacity of living organisms. Planarian flatworms are freshwater invertebrates that are rich in stem cells called neoblasts and, therefore, present a well-established model for studies on regeneration and the testing of novel antioxidant and radioprotective substances. In this work, we tested an antiviral and antioxidant drug Tameron (Monosodium α-Luminol or 5-amino-2,3-dihydro-1,4-phthalazinedione sodium salt) for its ability to reduce the harm of X-ray- and chemically induced oxidative stress on a planarian model. Our study has revealed the ability of Tameron to effectively protect planarians from oxidative stress while enhancing their regenerative capacity by modulating the expression of neoblast marker genes and NRF-2-controlled oxidative stress response genes.

## 1. Introduction

Antioxidants are compounds that reduce damage to molecules, organs, tissues, and the cells of a living organism caused by exposure to reactive oxygen species (ROS). Their effects are mostly based on boosting natural antioxidant cell defenses and ROS inactivation [1]. Antioxidants are highly important components of various therapeutic agents for oxidative stress is a common cause of many chronic and aging-related diseases [2]. A large number of antioxidants are also known as radioprotective substances, which can reduce the negative effects of ionizing radiation by neutralizing ROS that emerge as a result of X-ray interaction with water and biological molecules [3]. It is an established fact that the main damaging effect of ionizing radiation on living organisms is caused by the formation of different forms of free radicals [4]. On account of the presence of unpaired electrons, ROS possess a high level of redox activity, which causes oxidative damage in biological molecules such as nucleic acids, proteins, and membrane lipids [5].

The search for novel antioxidant and radioprotective substances for medical purposes is a task of high importance, despite the progress that has already been made. Radioprotectors are highly desired agents for radiation therapy [6,7,8,9]. Antioxidants may find their use in future therapies for aging and aging-related conditions, such as inflammaging, one of the crucial hallmarks of aging [10]. Amongst the different drug development strategies, repositioning existing drugs against new diseases is a promising approach as they have already passed preclinical and clinical tests and are well studied in terms of safety and side effects [11].

Tameron (Monosodium α-Luminol or 5-amino-2,3-dihydro-1,4-phthalazinedione sodium salt) is a novel generic drug also known as Galavit (GVT) or MP1032. This compound has already found its use as an antiviral and immunomodulatory agent utilized for the treatment of various conditions. It was applied as a neuroprotector against Gulf War disease, improving cognition, mood, and neurogenesis and alleviating neuro- and systemic inflammation [12]. It was also used against different viral infections, such as T cell-tropic, cytopathic retrovirus ts1 infection on a mouse model, protecting thymic epithelial cell cytoarchitecture and allowing thymocyte survival [13,14]. Its action mechanism involved the stabilization of the Nrf2 transcription factor, a known anti-inflammatory and antioxidant protein [15]. It was also tested on a mouse model and human peripheral blood mononuclear cells as an immunomodulatory agent for preventing cytokine storm, a devastating condition of the COVID-19 disease [16].

Given the effective results that were achieved by Tameron as an antiviral, immunomodulatory, and anti-inflammatory agent with an antioxidant mechanism of action, we decided to further explore its role as a potential antioxidant and radioprotective agent. For our tests, we have chosen planarians as a model.

Planarians are model animals with an extremely high regeneration ability because their bodies are extremely rich in neoblast stem cells (up to 30%) [17]. Neoblasts are totipotent stem cells that are capable of differentiating into all types of cells of the adult animal, including germline cells [18]. These cells give the planarian body an almost unlimited regenerative potential allowing it to replace missing body parts or even regenerate a whole worm from a small body fragment [19]. These features make planarians an excellent object for studies of regenerative processes, stem cell proliferation, testing of pharmacological substances, and aging in vivo [20]. In particular, the *Schmidtea mediterranea* planarian has many advantages over vertebrate models: the animals are cheap and easy to handle, their asexual strain is immortal and fast-reproducing, their genome is sequenced, and they are available in large amounts.

Planarian neoblasts, like any fast-proliferating cells, are extremely sensitive to ionizing radiation [21]. Doses of more than 15 Gy lead to complete neoblast death and termination of regenerative processes [22], which is followed by typical abdominal curling of the animals and further dying within several weeks after irradiation [23]. Smaller radiation doses kill only some parts of neoblasts, while the remaining parts are able to restore the stem cell population, providing the ability for normal regeneration [24]. Irradiation of planarians with sublethal X-ray doses, preserving a part of neoblasts with a potential for further regeneration, is the basis of our experimental model. The effect of potential radioprotectors can be easily quantified by measuring the regeneration rate of the blastema, the amount of surviving neoblasts, and the expression of neoblast marker genes as well as the activity of ROS generation in the planarian body. The same principles are applicable not only to ionizing radiation studies but also to studies of chemically induced oxidative stress. Our model has been previously proven to be robust and effective for testing new radioprotectors and antioxidants in the example of a classical antioxidant, N-acetylcysteine [25]. In the present work, we have tested the antioxidant, radioprotective, and pro-regenerative properties of Tameron on models of X-ray- and chemically induced oxidative stress in the *Schmidtea mediterranea* planarian.

## 2. Materials and Methods

### 2.1. Animals

An asexual strain of *Schmidtea mediterranea* flatworm (*Turbellaria, Platyhelminthes*) was cultured at room temperature, in darkened glass bowls containing a mixture of distilled and tap H_2_O at 1:2 vol. Animals were nourished twice a week with mosquitoes larvae (*Chironomidae*). Before the trials, planarians were starved for one week to exclude the possible interference of food components with the effects of X-ray treatment [26]. Then, animals with similar body lengths (about 8 mm) were chosen for the experiment. The planarians were decapitated, with the removal of circa 1/5 of the total body length containing the cephalic ganglion using a Carl Zeiss Stemi 2000 dissecting microscope with a thin eye scalpel. Before decapitation, the planarians were immobilized by placing them on a cooling table for 3–4 min. The number of animals included in each experimental group was the same (35 animals).

### 2.2. Computer-Assisted Morphometry In Vivo

The blastema regeneration rate was quantified using computer morphometry [27]. Seventy-two hours after decapitation, the planarians were photographed using a Carl Zeiss Stemi 2000 microscope equipped with a Carl Zeiss AxioCam camera (Appendix A). To calculate the blastema regeneration speed, the regeneration index *R = s/S* was used, where *s* is the blastema area and *S*—total body area. *S* and *s* values were determined using the Plana 4.0 software (author’s development). Thirty-five animals were used in each experimental or control group. The results demonstrated here are the mean values of three independent trials.

### 2.3. Whole-Mount Immunocytochemical Study of Planarian Stem Cell Mitotic Activity

Animals with a body length of about 4 mm were chosen for this study. The number of mitotic cells was calculated a week after decapitation. Animals were treated with a 7% solution of N-acetylcysteine for 5 min and fixed in 4% formaldehyde and 0.3% Triton X100 in PBS for 20 min. Staining for mitotic cells was conducted according to Newmark and Alvarado [28]. Briefly, we labeled a mitotic cell marker—phosphorylated histone H3—with a primary antibody (Santa Cruz, Dallas, TX, USA) at 1/1000 dilution and a CF488A-conjugated secondary antibody (Biotium, Fremont, CA, USA) in 1/1000 dilution. After washing them in PBS three times, the whole-mount specimens were placed in Vectashield Antifade Mounting Medium (Vector Labs, Burlingame, CA, USA) and analyzed using a Leica TCS SP5 confocal laser scanning microscope. The mitotic cell number and the mitotic index (amount of mitotic cells per 1 mm^2^ of planarian body) were then measured and calculated as described before [29,30]. The specificity of immunocytochemical staining was approved using a non-immune serum.

### 2.4. Experimental Testing Substances

Menadione (Sigma, Saint Louis, MO, USA) was diluted in DMSO to a stock concentration of 10^−3^ M and then diluted further to a working concentration of 10^−6^ M. The stock solution of Tameron (5-amino-2,3-dihydro-1,4-phthalazinedione sodium salt, ANO Engineering Physics Institute, Serpukhov, Russia) (Appendix A) was prepared by diluting it in milliQ water to 10^−2^ M, further diluting it to 10^−4^–10^−5^ M working concentrations.

### 2.5. Planarian X-ray Irradiation

The animals were irradiated using an X-ray device, RUT-12 (15 mA, 200 kV). For treatment, worms were placed into Petri dishes on filter paper moistened with water. In studies involving Tameron as an antioxidant or radioprotective agent, it was added 12 h before irradiation. The radiation doses were 10 and 15 Gy at a power of 2 Gy per min.

### 2.6. RT-PCR for Gene Expression Analysis

The state of the neoblast population was characterized by the expression of 13 neoblast marker genes (Appendix A) [31,32] and 23 oxidative stress response genes controlled by the NRF2 transcription factor (Appendix A) [33]. mRNA was isolated from five planarians with a magnetic beads isolation kit (Sileks, Moscow, Russia). The concentration of isolated mRNA was measured with Qubit RNA High Sensitivity (HS) Assay Kits (Thermo, Carlsbad, CA, USA) on the Qubit 4 device (Thermo, Carlsbad, CA, USA) and diluted to 10 ng/reaction for the reverse transcription reaction. Reverse transcription was carried out with a Sileks (Moscow, Russia) kit, using oligo dT primer. The outcoming cDNA served as a template for real-time PCR. The reaction was performed using a reaction mixture with SybrGreen (Evrogen, Moscow, Russia), on a QuantStudio 5 thermocycler (Thermo FS, Waltham, MA, USA). The level of gene transcription was normalized by two housekeeping genes Smed-ef1 (GenBank accession number AY067688) and Smed_01699 (GenBank accession number JX010505). A sample without the stage of reverse transcription served as a control for genomic DNA contamination and was amplified with genome-specific primers. Gene-specific primers were selected using the Primer Express program (Applied Biosystems, Waltham, MA, USA). The expression data were analyzed using the online service http://www.qiagen.com (accessed on 25 September 2022), the mayday-2.14 program (Center for Bioinformatics, Tübingen, Germany), and the Genesis program [34]. Only those results, of which changes in the gene expression level were observed at *p* < 0.05, were taken into account.

### 2.7. Cellular Antioxidant Activity (CAA) Assay in Planarians

ROS activity in the animal body after irradiation was identified using H2DCFDA (2,7-dichloro-dihydrofluorescein-diacetate-acetyl) or CellROX^®^ Green Reagent. This dye is a well-known fluorescent intracellular sensor of reactive oxygen species [35]. Planarians were put in a solution of 10 μM H2DCFDA (Biotium, Fremont, CA, USA) and incubated for 1 h in darkness. Next, the animals were treated with Tameron, washed twice with water, and irradiated using an X-ray device or incubated in menadione solution. The positive control group was obtained by pre-incubation of animals for 30 min in 100 μM H_2_O_2_ (Sigma, Saint Louis, MO, USA) or CellROX^®^ Green Reagent (C10444, Thermo, Carlsbad, CA, USA). Then, the planarians were anesthetized for 5–10 min in a 0.1% solution of chloretone (Sigma, Saint Louis, MO, USA) [36] and photographed with an Axio Scope A1 fluorescence microscope (Carl Zeiss, Jena, Germany) (Ex/Em = 492–495/517–527 nm). In the images obtained using the ImageJ program (National Institute of Health, Bethesda, MD, USA), the total fluorescence intensity of the animal body was evaluated. The measurement results were averaged over 15 animals.

### 2.8. Tameron-Mediated H2DCFDA Oxidation In Vitro

To study the possible catalysis of H2DCFDA-H_2_O_2_ oxidation by Tameron, we performed an in vitro study. Stock solution of H2DCFDA (2.5 mM) was prepared in DMSO. Tameron stock solution was prepared as described before. Stock solutions were further diluted in TE buffer (pH = 7.5). Experiment was performed on 5 groups of samples: 25 µM H2DCFDA, 25 µM H2DCFDA + 1 мM H_2_O_2_, 25 µM H2DCFDA + 1 мM H_2_O_2_ + 10^−3^ M Tameron, 25 µM H2DCFDA + 1 мM H_2_O_2_ + 10^−4^ M Tameron, and 25 µM H2DCFDA + 1 мM H_2_O_2_ + 10^−5^ M Tameron. The solutions were incubated for 30 min in the dark; then, fluorescence was measured with Qubit4 (Thermo FS, Waltham, MA, USA) fluorimeter in the channel Ex 430–495 nm, Em 510–580 nm. Every measurement was performed in 3 repeats.

### 2.9. Statistical Data Analysis

The data obtained were treated statistically by the Sigma-Plot 9.11 program (Systat Software Inc., Erkrath, Germany) using one-way ANOVA analyses of variance.

### 2.10. Ethical Standards

All procedures performed in this study involving animals were performed in accordance with the ethical standards of the institution at which the studies were conducted.

## 3. Results

### 3.1. Tameron Acts as Radioprotector, Boosting the Planarian Regeneration

Tameron has shown pro-regenerative properties in the model of regenerating planarian blastema. On the third and fourth days after decapitation, it enhanced the regeneration rate by a statistically significant difference compared to control animals. After irradiation with doses of 10 and 15 Gy, the regeneration capacity of planarians was reduced by 34.9% to 52.1% of normal value (*p* < 0.001), respectively, on the third day after irradiation, and by 34.8% и 49.8% (*p* < 0.001) on the fourth day. Tameron (10^−4^ M) was able to significantly increase the regeneration speed on the third day of regeneration by 25.2% for the 10 Gy dose, and by 56.6% for the 15 Gy dose (*p* < 0.001) (Figure 1). On the fourth day, after 10 and 15 Gy doses, the blastema area of Tameron-treated animals was 14% and 40.8% bigger, respectively, than the untreated control (*p* < 0.001). Tameron itself enhanced the blastema growth on the third and fourth days by 15.9% and 12%, respectively, in non-irradiated animals.

### 3.2. Tameron Increases the Survival of Mitotic-Active Planarian Neoblasts after X-ray Irradiation

It is known that planarian regeneration depends on neoblasts, which are capable of proliferating and differentiating into all types of planarian cells [18]. Being completely eliminated at doses higher than 30 Gy, they retain some viability and mitotic activity at sublethal doses of 10–15 Gy, which we have used in our work [25]. We could observe that sublethal X-ray doses drastically reduce the mitotic index (number of mitotic cells per 1 mm^2^) in planarians, but the presence of Tameron in a concentration of 10^−4^ M saves the phenotype, retaining 3 to 5 times more neoblasts on the first day after irradiation. On the third day, the results are less distinguishable, although we can still see a significant increase in the mitotic index after Tameron treatment at 15 Gy. The most pronounced effect could be observed at 15 Gy after 7 days, where the mitotic index of untreated planarians was close to zero, while in the Tameron-treated group, it was restored to 65.9% of the non-irradiated control group (Figure 2, Appendix A).

### 3.3. Tameron Saves the Expression of Neoblast Marker Genes after Sublethal Doses of X-ray Irradiation

To monitor the influence of Tameron on planarian regeneration activity more precisely, we have measured the expression of a panel of neoblast marker genes [32] by means of RT-qPCR. On the first day after irradiation, we could observe a clear downregulation of most neoblast markers (Figure 3). Treatment with Tameron, on the contrary, significantly upregulates all of the observed genes. The presence of Tameron partially ameliorates the effect of irradiation. On the first day, it boosts the expression level of part of the marker genes at 10 Gy while reducing the downregulation of the rest. At 15 Gy, the downregulation of all the genes is less pronounced after Tameron treatment. On the tenth day, the expression of most markers is still reduced at 10, and especially at 15 Gy; in the presence of Tameron, the expression of markers remains reduced at 10 Gy, but it is restored to control levels or even slightly higher at 15 Gy. In general, we can observe a protective effect of Tameron on the neoblast pool and stem cell proliferation, especially after higher doses of radiation.

### 3.4. Tameron Modulates the Expression of Oxidative Stress Response Genes

To monitor the antioxidant activity of Tameron in terms of influence on the expression pattern of genes responsible for planarian redox metabolism, we have analyzed a test panel of antioxidant-related genes by RT-qPCR (Figure 4). These genes are known to be controlled by the transcription factor NRF2 which is a therapeutic target of Tameron [14]. The expression was measured after 24, 48, and 72 h after 10 Gy irradiation in the presence and absence of Tameron and compared to unirradiated and untreated control. After 24 h, the expression of most analyzed genes was reduced compared to the control, but Tameron treatment upregulated or restored to normal levels a significant part of them. After 48 h, the expression of most of the analyzed genes was elevated in the 10 Gy-irradiated sample but slightly downregulated or brought to normal levels in Tameron-treated samples. These results may reflect the fact that Tameron leads to earlier induction of antioxidant genes, providing a prompter response to damage caused by ionizing radiation. After 72 h, expression of most antioxidant genes was brought to normal level or slightly downregulated in the 10 Gy-irradiated sample. In the Tameron-treated sample, we still observed the upregulation of six genes while other genes were downregulated, which reflects the continuation of the modulatory activity of Tameron on antioxidant genes.

### 3.5. Tameron Boosts Planarian Regeneration after Menadione-Mediated Inhibition

Menadione [37] is a prooxidant drug that is used in a classical model of oxidative stress induction in biological systems [38]. We have used it in our planarian regeneration test system in concentrations of 10^−6^ M to evaluate the antioxidant properties of Tameron. Menadione on its own significantly inhibited the planarian regeneration for 21.8%, while Tameron boosted the regeneration activity for 16.3% (10^−4^ M) and for 11.5% (10^−5^ M) (Figure 5). When Tameron was added to the menadione-treated animals in concentrations of 10^−4^ and 10^−5^ M, it significantly upregulated the regeneration speed, restoring it almost to normal values (Figure 5).

### 3.6. Tameron Modulates the Expression of Neoblast Marker Genes after Menadione-Induced Oxidative Stress in Intact and Regenerating Planarians

To further explore the pro-regenerative properties of Tameron, we tested its influence on the expression of neoblast marker genes in intact (Figure 6a) and regenerating (Figure 6b) planarians.

For intact animals (Figure 6a), Tameron treatment upregulates the expression of most genes from our panel (10 of 15) after 48 h—the highest upregulation was observed in the case of *smed-hnf-4* (22 times), *smed-nlk-1* (15 times), *smed-fgfr-1* (7 times), *smed-pbx-1* (7 times), and *smed-smad6/7* (10 times). After 72 h, 7 of 10 genes kept elevated levels of transcription. In the menadione-treated group, an upregulation boost could be observed already after 24 h of incubation for almost all tested genes, then followed by transcriptional downregulation of most genes after 48 h. Most pronounced downregulation was seen in the case of *smed-soxP-1*, *smed-soxP-2,* and *smed-soxP-3*. After 72 h, *smed-soxP-3* and *smed-soxP-1* were upregulated again, 101 and 106 times, respectively, compared to the second day of incubation. For planarians treated with the Tameron-menadione combination, upregulation of most genes after 24 h was observed as well. So, the transcription of *smed-hnf-4* was increased 11 times, *smed-pbx-1* 4 times, *smed-nlk-1* 4 times, and *smed-fgfr-4* times.

For regenerating planarians (Figure 6b), incubation with menadione gives only a modest stimulation of most test panel genes expression in 24 h, while *smed-soxP-2* and *smed-fgfr-4* are downregulated. On the second day, the expression of most genes is downregulated—for example, *smed-nlk-1*–15 times, *smed-soxP-1*, and *smed-soxP-2*–6 times. On the third day, the picture remains practically the same, except for 5 times downregulation of *smed-hnf-4* and 11 times upregulation of *smed-smad6/7*. After Tameron treatment of regenerating planarians, the picture is quite similar to that of menadione—the same genes were up- and downregulated on the second and third day of the experiment. The situation changed after combined menadione-Tameron treatment. On the second day of incubation, half of the genes in the panel were drastically upregulated: *smed-soxP-2*—3452 times, *smed-fgfr-4*–в—1942 times, *smed-soxP-1*—2004 times, *smed-nlk-1*—568 times, *smed-soxP-3*—1228 times, *smed-egr-1*—3219 times, *smed-zfp-1*—3121 times, and *smed-fgfr-1*—9630 times. Some of the genes were downregulated: *smed-prox-1* went down 48 times, and *smed-soxB-1* went down 23 times. However, on the third day, the upregulated half went down strongly, while the expression level of other genes returned to normal values.

### 3.7. Tameron Modulates the Expression of Oxidative Stress Response Genes after Menadione-Induced Oxidative Stress in Intact and Regenerating Planarians

On the first day after Tameron treatment of intact planarians (Figure 7a), we observed a pronounced downregulation of almost all genes of the panel, with the exception of *smed-GCL,* which went down during the following two days. On the second day, the previously downregulated *smed-gpx4, smed-alh1, smed-alh2, smed-prx6, smed-txn, smed-txnrd1, smed-sod1, smed-cat, smed-ugt2b7 smed-pgd, smed-taldo, smed-tkt,* and *smed-fth smed-9751* became activated. *Smed-txnrd1* expression was boosted 86 times, while all the others were upregulated 3–16 times. At the same time, the expression of *smed-GR* и *smed-G6PD* went down. On the third day, only the *smed-GR* gene was upregulated. A similar picture was observed in the group of animals incubated with Tameron and menadione: most parts of the genes downregulated on the first day were upregulated on the second day, and then the expression levels returned to normal values. A pronounced expression downregulation could be observed for the *smed-gpx4, smed-alh1*, and *smed-9751* genes. After treatment with menadione only, the same genes from the previous group were upregulated, but with several exceptions. The most pronounced upregulation was observed for the gene *smed-txnrd1* (22 times). On the third day, upregulation was observed for the *smed-pgd* and *smed-gr* genes.

In regenerating animals (Figure 7b), we could observe a significant upregulation of almost all oxidative stress response genes from the tested panel after menadione treatment on the first day of incubation. The only exception is the *Smed-gr2* gene—its activity was close to the control in all of the experimental groups. The biggest difference from the control was detected for the *Smed-tkt* and *Smed-txnrd1* genes (18 times upregulation). On the second day, the expression of *Smed-alh1*, *Smed-alh2*, *Smed-txnrd1*, *Smed-cat*, and *Smed-ptk1* genes remained elevated, while other genes became downregulated. The largest degree of downregulation could be observed for the *Smed-fth* gene (55 times). On the third day of incubation, the expression of the upregulated genes went down as well: 12 times for *Smed-alh1*, 8 times for *Smed-alh2*, 155 times for *Smed-txnrd1*, 29 times for *Smed-cat*, and 8 times for *Smed-ptk1*.

On the first day, after Tameron treatment, almost half of the panel was also upregulated. The highest overexpression was detected for the following genes: *Smed-tkt*, *Smed-fth*, *Smed-taldo*, and *Smed-pgd*; however, on the second day, their expression was downregulated by 7, 22, 7, and 9 times, respectively. The activity of catalase *Smed-cat* was elevated 7 times on the second day. On the third day of incubation with Tameron, the expression of the entire panel was inhibited. The activity of *Smed-txn* went down 46 times, *Smed-txnrd1*—61 times, and *Smed-alh2*—158 times.

After incubation of regenerating planarians with the Tameron-menadione combination, we observed upregulation of most genes on the first day; the most pronounced stimulation was detected for the *Smed-gr* gene (110 times) and *Smed-pgd* (7 times). On the second day, the expression of the following genes was further elevated: *Smed-cat* (3 times) and *Smed-alh2* (5 times). The *Smed-alh1* and *Smed-txnrd1* genes, which were downregulated on the first day, were upregulated on the second day. On the second day, the expression of the remaining genes was significantly reduced, particularly *Smed-gr*—691 times and *Smed-fth*—19 times. On the third day, the *Smed-alh2*, *Smed-cat*, and *Smed-txnrd1* were strongly downregulated, 392, 27, and 22 times, respectively. The expression of the remaining genes was insignificantly higher or lower than the control.

### 3.8. Tameron Is an Effective ROS Scavenger in the CAA Assay

To prove whether Tameron could quench ROS in situ, we performed a cellular antioxidant activity (CAA) assay with the reagent 2′,7′-Dichlorofluorescin diacetate (H2DCFDA), which is oxidized in the presence of ROS into a fluorescent dichlorofluorescein (DCF)—fluorescent intensity reflects ROS concentration in the cells of the studied specimen. Surprisingly, Tameron did not lower the DCF fluorescence, but, vice versa, enhanced it by itself, when combined with 10 and 15 Gy irradiation, which by itself predictably enhanced the fluorescence (Figure 8a). Menadione treatment boosted the fluorescence by itself, while the menadione-Tameron combination enhanced it even further on the first and second days and did not differ significantly from Tameron alone (Figure 8b). To clarify this paradoxical result, we incubated the dye H2DCFDA with H_2_O_2_ alone and with a combination of H_2_O_2_ and three gradually decreasing concentrations of Tameron, presuming that Tameron could directly catalyze the H2DCFDA oxidation, and thus produced an artifact in our experimental system rather than reflecting an increase in the ROS amount. Indeed, Tameron strongly boosted the oxidation of H2DCFDA in vitro, and this effect was concentration-dependent (Figure 8c). When we substituted H2DCFDA CellROX^®^ Green Reagent, we found out that Tameron did not promote CellROX^®^ Green Reagent fluorescence in an in vitro experiment. Using this reagent on ROS, we were able to detect the activity of Tameron as a free radical scavenger. As shown in Figure 8d, in the presence of exposure to ionizing radiation at doses of 5 and 10 Gy, the drug effectively reduced the amount of ROS almost to control values. Similarly, menadione increased the fluorescence intensity of CellROX^®^ by more than 2.5 times, and, in the presence of Tameron, the fluorescence intensity of the planarian body was comparable to that of the control group (Figure 8e). This indicates the effective scavenging of ROS by Tameron.

## 4. Discussion

Taking into account the chemical properties of Tameron, it is believed to scavenge abundant inflammation-induced ROS. During this process, Tameron is oxidized, generating a photon emission that can be detected by sensitive photon-counting devices—for example, by the ultra-weak photon emission (UPE) method [39]. Tameron has been described as an antiviral agent [16], boosting the cellular antioxidant stress defenses after viral infection by activation and stabilization of the redox-sensitive NF-E2-related transcription factor 2 (Nrf2) [13,15], which can ameliorate the devastating impact of immune reactions on viruses, including cytokine storm [40,41,42]. Similar effects were observed for Tameron on a model of Gulf War illness—a neurological condition induced by chemical damage to brain tissue [12].

Stabilization and activation of Nrf2 have previously been demonstrated to be a protective mechanism utilized by the endothelial cells against stress-induced oxidative damage [43], as well as in cadmium-induced oxidative stress [44]. In cells under the influence of oxidative stress, Nrf2 undergoes upregulation and translocation to the nucleus, where it upregulates the expression of genes related to GSH synthesis [45,46].

To prove whether Tameron is a potent antioxidant and radioprotector, we tested its action on our highly robust and reliable planarian model [25] for both X-ray- and chemically induced oxidative stress. In our study, we conducted a series of experiments to monitor the action of Tameron on intact and regenerating worms on multiple levels. These experiments are as follows: first, the measurements of the blastema regeneration rate; second, the calculations of the number of surviving neoblasts and an evaluation of their mitotic activity; third, the monitoring of the expression of the neoblast marker gene panel; fourth, the monitoring of the expression of antioxidant gene panel; and fifth, the estimation of ROS generation rate by the planarian tissue.

The experiments on the regeneration rate of X-ray-irradiated planarians and the neoblast mitotic activity have, indeed, confirmed that Tameron acts as a potent radioprotector. The studies on neoblast marker genes expression showed their clear upregulation after Tameron treatment on top of X-ray irradiation, giving clues to the mechanisms of the pro-regenerative and stem cell-protective activity of Tameron. The effect of Tameron on neoblast marker genes after chemically induced oxidative stress by menadione is more complex. We can still see that Tameron modulates the expression of neoblast marker genes when compared to untreated controls, thus influencing regenerative activity. In general, irradiation significantly lowers the expression levels of neoblast markers, which is most likely an outcome of neoblast death and results in the drastic collapse of the regeneration process. The presence of Tameron saves the expression of neoblast markers at levels comparable to unirradiated controls; this observed effect is a consequence of Tameron’s protective action. The drug protects planarian neoblasts and the regenerative potential of the animals. Moreover, Tameron itself is capable of stimulating the neoblast marker expression and thereby stimulating the regenerative process itself.

Concerning the NRF2-controlled genes responsible for oxidative stress response genes, the X-ray irradiation did not lead, in general, to a significant induction of their expression. Still, we have to point out that we have observed an increase in the expression of the *alh* gene responsible for the oxidation of aldehyde and ketone groups which are formed in lipids as a result of ionizing radiation [47]. The presence of Tameron also increased the expression of this gene after irradiation, while the expression of glutathioneperoxidase was significantly reduced three days after irradiation, which may indicate that Tameron is capable of efficiently degrading the peroxide compounds by itself, thereby reducing the levels of oxidative stress.

In cases of menadione-induced oxidative stress, Tameron also boosts the regenerative activity of planarians, demonstrating its antioxidant potential. On the level of antioxidant response genes, it has a complex influence in cases of X-ray irradiation, strongly upregulating some of them. In the case of menadione-induced oxidative stress, Tameron also produces a complex pattern of changes in the antioxidant response gene expression panel, upregulating some of them, which may be crucial for combating oxidative stress. As we have shown, menadione in intact planarians significantly stimulated the level of expression of neoblast marker genes. It is likely that oxidative stress caused by the presence of menadione is a trigger mechanism for activating the regenerative activity of stem cells. It was previously demonstrated that one of the triggers for the activation of neoblast proliferation and regeneration is ROS activity in the area of damage to the animal body [48]. Obviously, in regenerating animals, the mechanism of neoblast activation has already been launched, and an additional stimulus in the form of ROS from menadione leads, on the contrary, to a decrease in the expression of neoblast gene markers, which is obviously a consequence of the suppression of the regenerative process by menadione in planarians. At the same time, as we have already said, Tameron by itself activates the regeneration of planarians, which manifests itself at elevated levels of expression of neoblast markers (an active proliferative process is underway) in intact animals, while in regenerating animals, the inhibitory effect of menadione is neutralized by Tameron, which can be traced by the expression of neoblast markers—for example, after 24 h of regeneration, it is mostly similar to the control values.

Oxidative stress caused by menadione significantly increased the transcription of genes controlled by NRF2 in planarians. In intact planarians, most of the studied genes were actively transcribed after 48 h, and in regenerating ones after 24 h of incubation. These differences in dynamics are probably related to the physiological status of the animals—as we said above, regenerating planarians already have an increased level of ROS and, accordingly, menadione enhances this effect immediately after being added. In intact animals, the level of ROS increases after the addition of menadione, and this, accordingly, causes additional activation of the antioxidant defense system as early as 24 h of incubation. Tameron, being an NRF2 stabilizer, activates the antioxidant defense system itself (as can be seen from the level of gene expression that is observed in intact animals at 48 h of incubation and in regenerating animals at 24 h) due to this factor. Moreover, the differences in dynamics are, again, associated with the physiological state of the animals and the initial level of ROS. Furthermore, the effect of the combination of Tameron and menadione on the level of expression of antioxidant defense genes controlled by NRF2 follows the same dynamics, as is observed when these substances are exposed separately from each other. Additionally, since both of these substances activate antioxidant defense systems in almost the same way in dynamics and patterns, their combined effects are also similar on the level of gene transcription.

Experiments on direct measurement of ROS using the H2DCFDA oxidation to DCF did not provide reliable results in our hands. As previously reported [49], DCF can actually produce O_2_^•−^ and H_2_O_2_ via the reaction of DCF radical with oxygen, thus artificially elevating the very ROS that it is attempting to quantify. That was proven in our assay, creating an artifact, and making it unable to precisely quantify the ROS amount after X-ray- and menadione-induced oxidative stress and Tameron treatment. Indeed, Tameron enhances the oxidation of H2DCFDA and, accordingly, fluorescence is evidence of this. Obviously, Tameron is involved in the redox process, and this gives an inverse relationship in terms of the level of H2DCFDA fluorescence. While neutralizing free radicals, Tameron increases the intensity of H2DCFDA fluorescence. In the case of using CellROX^®^ Green, we, indeed, observed the effect of Tameron as a free radical scavenger, as this reagent does not get oxidized by Tameron. This should be taken into account when using the antioxidant properties of other substances—the absence of ROS scavenging effect may be an artifact due to the chemical interaction of the dye and the substance studied.

## 5. Conclusions

In general, Tameron has shown itself to be a potent novel radioprotector and antioxidant, robustly enhancing the regeneration rate in planarians damaged by X-ray- or menadione-induced oxidative stress and protecting mitotic activity of neoblasts while modulating the expression patterns of neoblast markers and oxidative stress response genes.

## Figures and Tables

**Figure 1 antioxidants-12-00953-f001:**
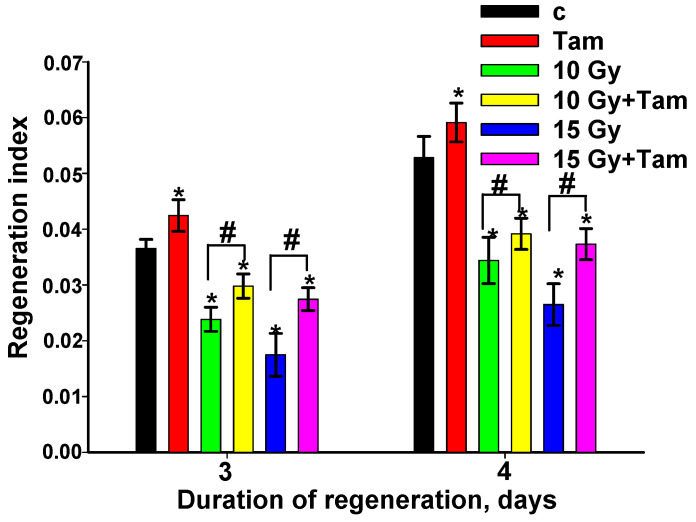
Radioprotective effects of Tameron on regenerating planarians in non-irradiated animals and after X-ray irradiation with 10 and 15 Gy. c—non-irradiated control, Tam—Tameron (10^−4^ M). * *p* < 0.001 (difference from control), ^#^
*p* < 0.001 (difference from irradiated group without Tameron treatment). Data are presented as mean  ±  SD.

**Figure 2 antioxidants-12-00953-f002:**
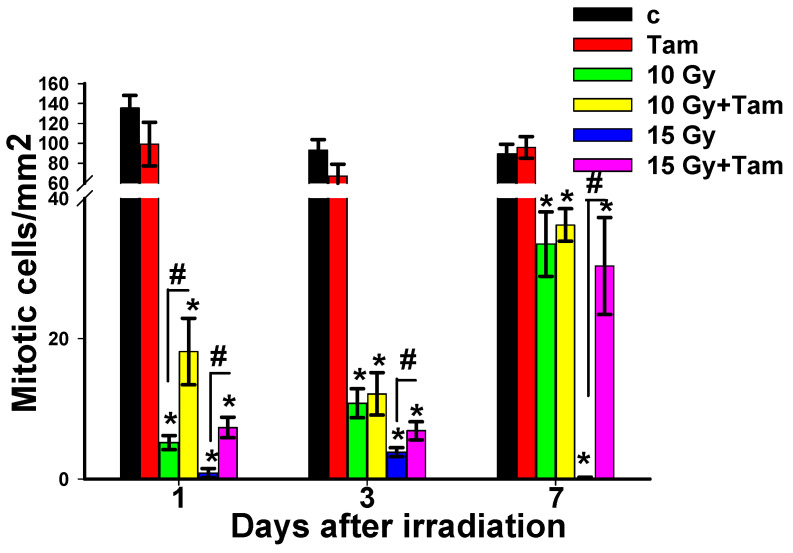
Tameron protects proliferating planarian neoblasts from elimination after X-ray irradiation at sublethal doses of 10 and 15 Gy. c—non-irradiated control, Tam—Tameron (10^−4^ M). * *p* < 0.001 (difference from control), ^#^
*p* < 0.001 (difference from irradiated group without Tameron treatment). Data are presented as mean  ±  SD.

**Figure 3 antioxidants-12-00953-f003:**
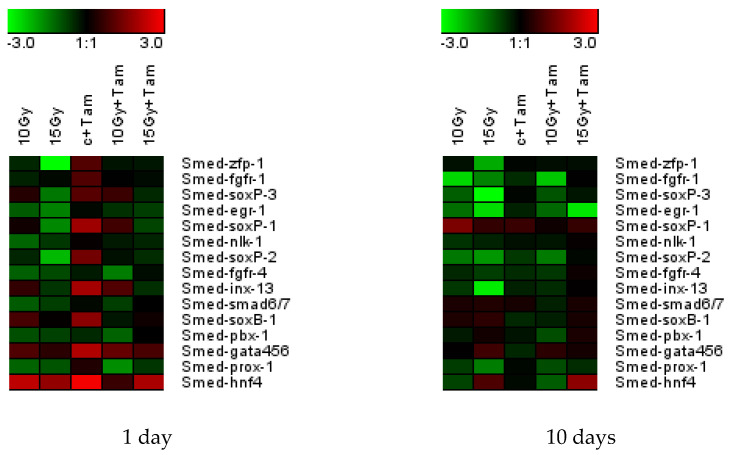
Tameron stimulates expression of neoblast marker genes. The figure shows a heatmap reflecting changes in expression of a test panel of neoblast marker genes in planarians after irradiation and/or treatment with Tameron (Tam) (10^−4^ M) compared to unirradiated and untreated animals. The scale of intensity of the standardized expression values extends from −3 (green: low expression) to +3 (red: high expression). The 1:1 intensity value (black) represents the non-treated control. The measurements were made on the first and on the tenth day after irradiation.

**Figure 4 antioxidants-12-00953-f004:**
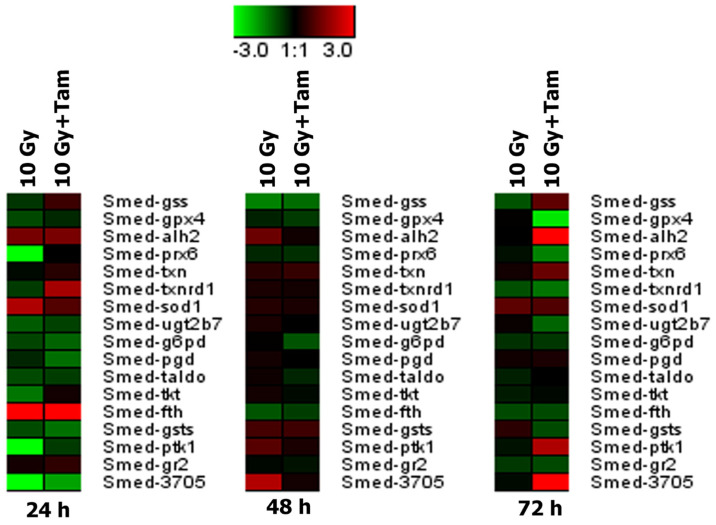
Tameron modulates expression of oxidative stress response genes. The figure shows a heatmap reflecting changes in expression of a test panel of NRF-2-controlled antioxidant defense genes in planarians after irradiation and/or treatment with Tameron (Tam) (10^−4^ M) compared to unirradiated and untreated animals. The scale of intensity of the standardized expression values extends from −3 (green: low expression) to +3 (red: high expression). The 1:1 intensity value (black) represents the non-treated control.

**Figure 5 antioxidants-12-00953-f005:**
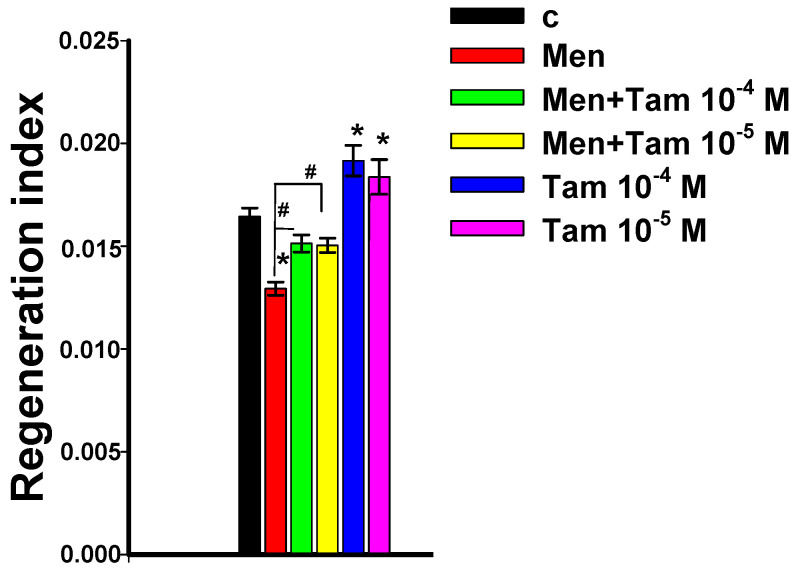
Antioxidant effects of Tameron on regenerating planarians in non-treated animals and after menadione-induced oxidative stress. c—non-irradiated control, Tam—Tameron, Men—menadione. * *p* < 0.001 (difference from control), ^#^
*p* < 0.001 (difference from irradiated group without Tameron treatment). Data are presented as mean  ±  SD.

**Figure 6 antioxidants-12-00953-f006:**
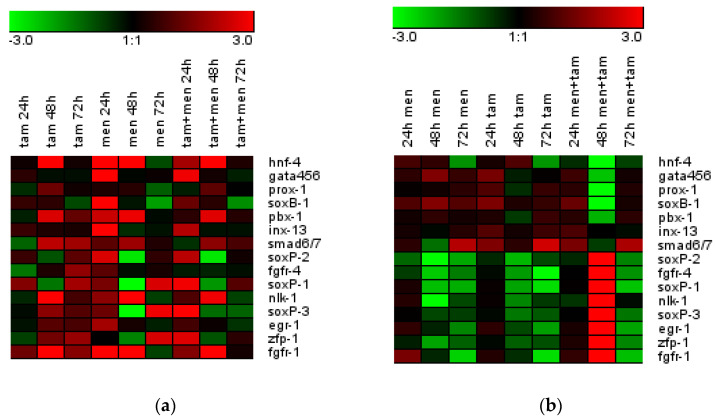
Tameron effect on expression of neoblast marker genes in intact (**a**) and regenerating (**b**) planarians. The figure shows heatmaps reflecting changes in expression of a test panel of neoblast marker genes in planarians after menadione-induced oxidative stress and/or treatment with Tameron compared to untreated animals. The scale of intensity of the standardized expression values extends from −3 (green: low expression) to +3 (red: high expression). The 1:1 intensity value (black) represents the non-treated control. A non-treated control group was taken as a control. Menadione (Men) was used at working concentration of 10^−6^ M, Tameron (Tam)—10^−4^ M. The measurements were made on the 24th, 48th, and 72nd h after treatment.

**Figure 7 antioxidants-12-00953-f007:**
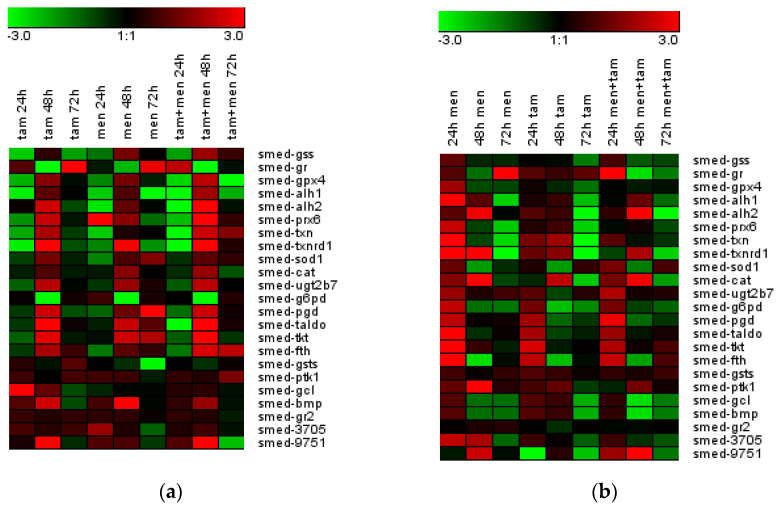
Tameron’s effect on expression of oxidative stress response genes in intact (**a**) and regenerating (**b**) planarians. The figure shows heatmaps reflecting changes in expression of a test panel of neoblast marker genes in planarians after menadione-induced oxidative stress and/or treatment with Tameron compared to untreated animals. The scale of intensity of the standardized expression values extends from −3 (green: low expression) to +3 (red: high expression). The 1:1 intensity value (black) represents the non-treated control. A non-treated control group was taken as a control. Menadione (Men) was used at working concentration of 10^−6^ M, Tameron (Tam)—10^−4^ M. The measurements were made on the 24th, 48th, and 72nd h after treatment.

**Figure 8 antioxidants-12-00953-f008:**
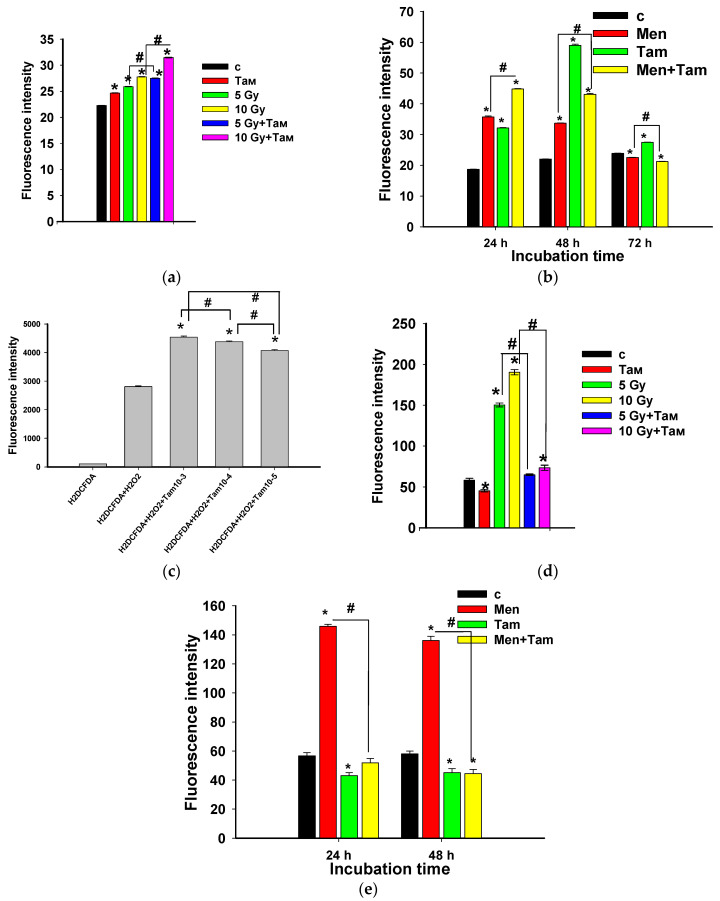
Tameron-mediated catalysis of H2DCFDA oxidation in vivo and in vitro. (**a**) Effect of Tameron (Tam) on DCF fluorescence in intact and 5 and 10 Gy-irradiated worms. (**b**) Effect of Tameron, menadione (Men), and Tameron-menadione combination on DCF fluorescence in planarians after 24, 48, and 72 h. (**c**) Tameron catalyzes H2DCFDA oxidation by H_2_O_2_ in a concentration-dependent manner. (**d**) The effect of Tameron as a ROS scavenger in planarian bodies after X-ray exposure, revealed with CellROX^®^ Green dye. (**e**) Effect of Tameron as a ROS scavenger in planarian bodies after menadione exposure, revealed using CellROX^®^ Green dye. c—non-treated control, Tam—Tameron, Men—menadione. * *p* < 0.001 (difference from control), ^#^
*p* < 0.001 (difference from group without Tameron treatment). Data are presented as mean  ±  sem.

## Data Availability

The data are contained within this article.

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
