# Peer review of "A Study on the Planarian Model Confirms the Antioxidant Properties of Tameron against X-ray- and Menadione-Induced Oxidative Stress"

_antioxidants, 2023, doi:10.3390/antiox12040953_

Round 1

Reviewer 1 Report

The paper entitled “A study on planarian model confirms antioxidant properties of 2 Tameron against X-ray- and menadione-induced oxidative 3 stress 4” offers a wide analysis on the antioxidant properties of tameron in an in vivo model system in which different aspects of stem cell regulation, proliferation and differentiation can be monitored at the same time. Authors used both chemical and X-ray induced oxidative stress thus providing a comprehensive experimental context. However, some issues have to be fixed to make it suitable for publication in Antioxidant journal.

1) most of the experiments were performed by using Real-time RT-PCR assays. Thus the choice of the reference genes is a critical point. The authors used two reference housekeeping genes and did not provided literature reference about their stability following X-ray treatment and in chemical induced oxidative stress. This makes the results obtained by Real time RT-PCR experiments weak. Authors should provide a statistical evaluation of reference genes stability across their experimental classes or, alternatively demonstrate the stability of reference genes by using a different quantification assay such as for example densitometry of hybridization signal in whole mount in situ hybridization experiments.  

2) for immunolocalization experiments, some representative images, of demonstrating staining quality, should be included. The same for morphometric analysis of blastema size.

3) the paragraph “3.8. Tameron does not Lower the Fluorescence of ROS-Indicator DCF in the CAA Assay” is useless as the same authors demonstrate a technical artifact. The authors might try to test the effect of tameron on ROS production by replacing H2DCFDA with a different reagent. In line with the manuscript setup and the journal in which it is submitted for publication, this attempt might be relevant.

4) the authors present several data about gene expression changes at different time and in different experimental conditions showing a variegate fluctuation of different markers making the reading of paragraphs 3.6 and 3.7 very difficult. At least at the end, the reader expects a discussion about these fluctuations which the authors don't actually address! Thus making the discussion in its actual form very superficial.

5) There are numerous typos and grammatical errors in the manuscript that need to be corrected

Author Response

We are very grateful for the reviewers’ comments that are aimed at improving our paper. We have thoroughly revised the manuscript in accordance with the reviewers’ comments. We have carefully checked all the points and we have tried to address all the questions and suggestions.

Reviewer #1

General issue: The paper entitled “A study on planarian model confirms antioxidant properties of 2 Tameron against X-ray- and menadione-induced oxidative 3 stress 4” offers a wide analysis on the antioxidant properties of tameron in an in vivo model system in which different aspects of stem cell regulation, proliferation and differentiation can be monitored at the same time. Authors used both chemical and X-ray induced oxidative stress thus providing a comprehensive experimental context. However, some issues have to be fixed to make it suitable for publication in Antioxidant journal.

Discussion: We thank the reviewer for the positive evaluation of our work.

All corrections in the article text are marked in yellow

Issue 1:  most of the experiments were performed by using Real-time RT-PCR assays. Thus the choice of the reference genes is a critical point. The authors used two reference housekeeping genes and did not provided literature reference about their stability following X-ray treatment and in chemical induced oxidative stress. This makes the results obtained by Real time RT-PCR experiments weak. Authors should provide a statistical evaluation of reference genes stability across their experimental classes or, alternatively demonstrate the stability of reference genes by using a different quantification assay such as for example densitometry of hybridization signal in whole mount in situ hybridization experiments.

Discussion: In the experiments, the same amount of animals of approximately the same size was taken for each experimental group and series. It allows to isolate a similar amount of mRNA from each sample to evaluate the levels of gene expression. (we have additionaly measured the isolated mRNA concentration in the samples by means of the Qubit 4 device supplied with the Qubit™ RNA High Sensitivity (HS) Assay Kits). In case of significant concentration aberrations the mRNA concentration was brought to 10 ng/reaction prior to cDNA synthesis. The Ct level evaluation from group to group demonstrated an acceptable level of stability, which did not depend on type of impact, what shows us an acceptable level of applicability of this genes as housekeeping reference genes (Table 1). This housekeeping genes are widely used in gene expression studies in planarians. (Ramirez, A.N.; Loubet-Senear, K.; Srivastava, M. A Regulatory Program for Initiation of Wnt Signaling during Posterior Regeneration. Cell reports 2020, 32, 108098, doi:https://doi.org/10.1016/j.celrep.2020.108098; Yuwen, Yq., Dong, Zm., Wang, Qh. et al. Evaluation of endogenous reference genes for analysis of gene expression with real-time RT-PCR during planarian regeneration. Mol Biol Rep 38, 4423–4428 (2011). https://doi.org/10.1007/s11033-010-0570-8; Wenemoser D, Lapan SW, Wilkinson AW, Bell GW, Reddien PW. A molecular wound response program associated with regeneration initiation in planarians. Genes Dev. 2012 May 1;26(9):988-1002. doi: 10.1101/gad.187377.112. PMID: 22549959; PMCID: PMC3347795).

Table 1 Ct values for the housekeeping genes in experiments after Tameron treatment and X-ray irradiation.

Ct

housekeeping genes

сontrol

10Gy

15Gy

tam

10Gy+tam

15Gy+tam

Mean

St.dev

Smed-ef1

repeat 1

13,53682

13,89034

13,53018

13,23888

13,83071

13,59679

13,60395

0,235251

repeat 2

13,37982

13,65368

13,43206

13,56614

13,73182

13,74809

13,58527

0,154091

repeat 3

13,46491

13,45386

13,25081

13,63696

13,70096

13,37257

13,48001

0,166398

Smed_01699

repeat 1

21,10318

21,05798

21,65507

21,17804

21,22817

21,54419

21,29444

0,246113

repeat 2

21,40535

21,77964

21,58871

21,56395

21,4088

21,78737

21,58897

0,168832

repeat 3

21,37394

21,00893

21,10346

21,49975

21,2695

21,50101

21,29277

0,204811

Changes made in the manuscript: Additional data were introduced into the Methods section specifying the mRNA concentration measurement method and its normalization before the reverse transcription reaction (Section 2.6).

Issue 2 for immunolocalization experiments, some representative images, of demonstrating staining quality, should be included. The same for morphometric analysis of blastema size.

Changes made in the manuscript: The Figire S1 has been added to the Supplementary materials with representative pictures of regenerating planarians and their blasthema along with Figure S3 with pictures of the animals after immunohistochemical staining of the mitotic cells.

Issue 3 the paragraph “3.8. Tameron does not Lower the Fluorescence of ROS-Indicator DCF in the CAA Assay” is useless as the same authors demonstrate a technical artifact. The authors might try to test the effect of tameron on ROS production by replacing H2DCFDA with a different reagent. In line with the manuscript setup and the journal in which it is submitted for publication, this attempt might be relevant.

Discussion: Indeed, we got an apparently opposite effect due to ability of Tameron to directly react with H2DCFDA. In the Discussion session we propose that other reagents exist capable of producing such artifacts due to their antioxidant properties and direct interaction with H2DCFDA, what may lead to false statement about absence of antioxidant properties. We have conducted additional research with using the CellROX reagent as an alternative. This fluorescent ROS indicator did not react with Tameron and allowed us to obtain reliable data on antioxidant properties of Tameron.

Changes made in the manuscript: Changes have been made and data have been added to the sections 2.7 and 3.8. (the method description and the results obtained with the CellROX dye) and corresponding part has been added to the discussion section.

Issue 4 the authors present several data about gene expression changes at different time and in different experimental conditions showing a variegate fluctuation of different markers making the reading of paragraphs 3.6 and 3.7 very difficult. At least at the end, the reader expects a discussion about these fluctuations which the authors don't actually address! Thus making the discussion in its actual form very superficial.

Discussion: We have added a discussion of relationship between the changes in gene expression and observed macroeffects.

«The experiments on regeneration rate of X-ray irradiated planarians and on the neoblast mitotic activity have indeed confirmed that Tameron acts as a potent radioprotector. The studies on neoblast marker genes expression showed their clear upregulation after Tameron treatment on top of X-ray irradiation, giving clues to mechanisms of pro-regenerative and stem cell-protective activity of Tameron. The effect of Tameron on neoblast marker genes after chemically induced oxidative stress by menadione is more complex. We still can see that Tameron modulates expression of neoblast marker genes when compared to untreated controls, thus influencing the re-generative activity. In general, the irradiation significantly lowers the expression levels of neoblast markers, what is most possibly an outcome of neoblast death and results in drastic collapse of the regeneration process. The presence of Tameron saves the expres-sion of neoblast markers at levels comparable to unirradiated controls, this observed effect is a consequence of Tameron protective action. The drug protects planarian neoblasts and the regenerative potential of the animals. Moreover, Tameron itself is capable of stimulating the neoblast marker expression and thereby – of stimulating the regenerative process itself.

Concerning the NRF2-controlled genes responsible for oxidative stress response genes, the X-ray irradiation did not lead in general to significant induction of their ex-pression. Still, we have to point out that we have observed an increase in expression of the alh gene responsible for oxidation of aldehyde and ketone groups which are formed in lipids as a result of ionizing radiation [47]. The presence of Tameron aditionally in-creased the expression of this gene after irradiation, while the expression of glutathioneperoxidase was significantly reduced three days after irradiation what may indicate that Tameron is capable of efficiently degrading the peroxide compounds by itself, thereby reducing the levels of oxidative stress.

In case of menadione-induced oxidative stress, Tameron as well boosts regenerative activity of planarians, demonstrating its antioxidant potential. On the level of antioxidant response genes it has a complex influence in case of X-ray irradiation, strongly upregulating some of them. In case of menadione-induced oxidative stress, Tameron also produces a complex pattern of changes in the antioxidant response gene expression panel, upregulating some of them which may be crucial for combating oxidative stress. As we have shown, menadione in intact planarians significantly stimulated the level of expression of neoblast marker genes. It is likely that oxidative stress caused by the presence of menadione is a trigger mechanism for activating the regenerative activity of stem cells. So it was previously demonstrated that one of the triggers for the activation of neoblast proliferation and regeneration is ROS activity in the area of damage to the animal body [48]. Obviously, in regenerating animals, the mechanism of neoblast activation has already been launched, and an additional stimulus in the form of ROS from menadione leads, on the contrary, to a decrease in the expression of neoblast gene markers, which is obviously a consequence of the suppression of the regenerative process by menadione in planarians. At the same time, as we have already said, Tameron by itself activates the regeneration of planarians, which manifests itself at elevated levels of expression of neoblast markers (an active proliferative process is underway) in intact animals, while in regenerating animals the inhibitory effect of menadione is neutralized by Tameron, what can be traced by the expression of neoblast markers - for example, on 24 hours of regeneration, it is mainly similar to the control values.

Oxidative stress caused by menadione in planarians significantly increased the transcription of genes controlled by NRF2. In intact planarians, most of the studied genes were actively transcribed after 48 hours, and in regenerating ones, after 24 hours of incubation. These differences in dynamics are probably related to the physiological status of the animals - as we said above, regenerating planarians already have an increased level of ROS and, accordingly, menadione enhances this effect immediately after addition. In intact animals, the level of ROS increases already after the addition of menadione, and this, accordingly, causes additional activation of the antioxidant defense system as early as 24 hours of incubation. Tameron, being a NRF2 stabilizer, activates the antioxidant defense system itself (as can be seen from the level of gene expression that is observed in intact animals at 48 hours of incubation, and in regenerating animals at 24 hours) due to this factor. Moreover, the differences in dynamics are again associated with the physiological state of the animals and the initial level of ROS. And the effect of the combination of Tameron and menadione on the level of expression of antioxidant defense genes controlled by NRF follows the same dynamics as is observed when these substances are exposed separately from each other. And since both of these substances activate antioxidant defense systems in almost the same way in dynamics and patterns, their combined effects are also similar on the level of gene transcription.»

Changes made in the manuscript: The information presented above has been added to the Discussion section clarifying the relationship between the changes in gene expression and the observed macroeffects – namely oxidative stress development, neoblast proliferation processes, regeneration and the oxidative stress.

Issue 5 There are numerous typos and grammatical errors in the manuscript that need to be corrected

Discussion: We thank the reviewer for this correction.

Changes made in the manuscript: The manuscript has been revised one more time for typos and grammar mistakes.

Reviewer 2 Report

The authors are performing radio-protecting effect of an antiviral drug, Tameron, using planarian model. This reviewer has some questions below.

Tameron is not so popular drug. Please include the chemical structure of the Tameron.

What is the target ROS in the planarian irradiated by X-ray? Is it H2O2?

The horizontal axis of Figure 2 lacks some numbers.

This reviewer can not understand why H2DCFDA assay was performed in this study. The H2DCFDA assay did not support the role of Tameron as ROS-scavenger. Please do suitable ROS measurement.

Author Response

We are very grateful for the reviewers’ comments that are aimed at improving our paper. We have thoroughly revised the manuscript in accordance with the reviewers’ comments. We have carefully checked all the points and we have tried to address all the questions and suggestions.

General issue: The authors are performing radio-protecting effect of an antiviral drug, Tameron, using planarian model. This reviewer has some questions below.

Discussion: We thank the reviewer for the positive evaluation of our work.

All corrections in the article text are marked in yellow

Issue 1 Tameron is not so popular drug. Please include the chemical structure of the Tameron.

Changes made in the manuscript: In the supplementary materials we have presented the structural formula of Tameron on Figure S2.

Issue 2 What is the target ROS in the planarian irradiated by X-ray? Is it H2O2?

Discussion: According to existing data, the X-ray irradiation first of all causes the water radyolisis, which is accompanied by formation of free radicals, namely HO∙, H∙, HO2∙, H3O+, OH−, and the hydrogen peroxide H2O2 (Spinks, J.W.T.; Woods, R.J. An Introduction to Radiation Chemistry, 3rd ed.; Wiley-Interscience publication: New York, NY, USA, 1990; Ferradini, C.; Jay-Gerin, J.P. The effect of pH on water radiolysis: a still open question. A minireview. Res. Chem. Intermed. 2000, 26, 549–565; Le Caër, S. Water Radiolysis: Influence of Oxide Surfaces on H2 Production under Ionizing Radiation. Water 2011, 3, 235-253. https://doi.org/10.3390/w3010235)

Issue 3 The horizontal axis of Figure 2 lacks some numbers.

Changes made in the manuscript: Corrected. The numbers were added.

Issue 4 This reviewer can not understand why H2DCFDA assay was performed in this study. The H2DCFDA assay did not support the role of Tameron as ROS-scavenger. Please do suitable ROS measurement.

Discussion: Indeed, we got an apparently opposite effect due to ability of Tameron to directly react with H2DCFDA. In the Discussion session we propose that other reagents exist capable of producing such artifacts due to their antioxidant properties and direct interaction with H2DCFDA, what may lead to false statement about absence of antioxidant properties. We have conducted additional research with using the CellROX reagent as an alternative. This fluorescent ROS indicator did not react with Tameron and allowed us to obtain reliable data on antioxidant properties of Tameron.

Changes made in the manuscript: Changes have been made and data have been added to the sections 2.7 and 3.8. (the method description and the results obtained with the CellROX dye) and corresponding part has been added to the discussion section.

Round 2

Reviewer 1 Report

x